# The Review of New Scientific Developments in Drilling in Wood-Based Panels with Particular Emphasis on the Latest Research Trends in Drill Condition Monitoring

**Jarosław Górski** 

Institute of Wood Sciences and Furniture, Warsaw University of Life Sciences (WULS), 166 Nowoursynowska St., 02-787 Warsaw, Poland; jaroslaw_gorski@sggw.edu.pl

**Abstract:** Drilling is still one of the basic cutting processes that are of particular interest to wood science and technology professionals. As a result, considerable (and very diverse thematically) research has been recently carried out on drilling. The article focuses on the new and quite spectacular approach to drill condition monitoring in wood-based panels machining. One of the most innovative elements in the analyzed research trend is the adoption of the new general methodological assumptions that allow one to define the drill condition monitoring problem as a standard three-class classification. The general effectiveness of the tested monitoring systems (accuracy of classification ACC [%]), ranged between 67% and 82%. The critical classification error (CCE [%]) ranged between 0% and 1.6%. These results seem very promising, yet are still not good enough to develop a commercial monitoring system. A more useful form of obtaining diagnostic data and more effective classification strategies (algorithms) are likely to be required.

**Keywords:** tool condition monitoring; drilling; artificial intelligence



## 1. Introduction

The technique of wood drilling has a long history that dates to the Upper Paleolithic Period (about 40,000 years ago) [1] with numerous developments made in this field during these centuries/millennia [2]. Drilling is still one of the basic cutting processes that are of particular interest to wood science and technology professionals. It is in fact, from a practical standpoint, quite impossible to imagine the manufacturing of furniture and other interior elements or construction of wooden buildings without effective drilling techniques. It is also worth noting that drilling can be used for more than just making holes in wood (or wood-based boards) for construction purposes. There are many other areas in which drilling can be very useful from a practical point of view (i.e., in semi-destructive testing of various physical and technological properties of wood or wood-based materials). As a result, considerable (and very diverse thematically) research has been recently carried out on drilling. This ensures constant development—new research trends constantly appear in scientific journals. The fundamental and traditional research direction (which resulted in a significant development of practical knowledge), was to study the problem of increasing the hole quality. These studies most often focused on the delamination factor [3] and surface roughness [4,5] in wood-based panel drilling. Moreover, the drill deflection problem (which is the main cause of the hole position error) has recently been analyzed in an innovative way [6–8].

Another traditional research direction was the study of cutting forces generated during drilling [9,10]. In parallel, new research on improving the drilling resistance measurements as a semi-destructive method of assessing the condition of the existing timber structures are still being carried out [11–13]. Likewise, a method of estimation of wood density and/or moisture content by drilling chips extraction technique is constantly being improved [14–17]. Similarly, the advanced scientific studies on the development of drilling-based tests (which are increasingly and commonly considered the most convenient—the quickest and most

material-saving—methods of machinability rating of any wood-based materials) has been carried out recently [18–20].

A different, and also particularly important, research trend concerns the problem of drill bits wear. The issue of drill wear is related to the aforementioned hole quality be-cause, in general, there is an obvious relationship between tool wear and the machining quality. From a practical standpoint, we can do two things—extend the tool life (if possible) and replace the tool with a new one at the right moment (not too late and not too early). For this very reason, for example, some innovative types of anti-wear coating on the drill cutting blades are being tried out [21]. At the same time, however, the problem of drill condition monitoring should be solved. This kind of monitoring can be carried out by a machine operator, or the tool can be automatically replaced in specified intervals (e.g., every 4 h, or after a certain batch of workpieces). The first variant requires the employment of experienced workers and thus, contradicts the idea of full automation of machine tools. The second variant is not very effective due to the random variability of tool life. It is no wonder then that drill condition monitoring systems are an important topic in every industry. Therefore, these systems have been and are still being investigated by many research teams around the world [22,23]. Generally, tool condition monitoring in the field of woodworking has also been popular for a long time [24–26]. Therefore, at the end of this introductory (and as concisely as possible) overview of the latest research trends, it is also worth noting the new and quite spectacular approach to drill condition monitoring in wood-based panels machining [27–38].

Virtually all the mentioned above research trends deserve to be reviewed and analyzed, however it would go beyond the scope of a single article. Therefore, the article focuses only on the last one of the mentioned topics. In the last 2 or 3 years, numerous research and scientific analyses have been carried out to address this problem. Some of them seem to be truly innovative and their synthetic review is the main content of this paper [33–38]. The purpose is to encourage more scientists to cooperate or compete in this research area (both attitudes are beneficial from a scientific development point of view) currently. It would be useful since the tool condition monitoring is an important issue that needs to be solved for the sake of the furniture industry—preferably as soon as possible. Currently, there is no commercial (or even prototypical, for that matter) drill condition monitoring system that would make a serious offer for the wood or furniture industry.

At this point it is worth explaining why tool condition monitoring systems are extremely important for furniture factories. This is a natural consequence of global phenomena that go far beyond furniture manufacturing. Many industrialized countries face a progressing demographic crisis, which results in (among a plethora of other issues) a continuously decreasing number of employees ready to take up monotonous and low-paid work on production lines. Thus, the automation of production processes seems to be an absolute priority. The limitation of direct human participation in the furniture manufacturing processes can also bring many benefits, and not just economic ones. In this context, the term "Industry 4.0" (which describes the latest trend of technical development that involves intelligent automation in manufacturing technologies) has been essential for quite some time in every area of the industry, including the wood industry [39]. Modern furniture manufacturers have mainly used automated machine tools. This is true even for relatively small factories (although, of course, not in carpentry workshops). Most often, computer numerical control (CNC) systems are adopted for the automation of many machining tools, such as CNC drilling machines. It has been well known for several years that this is beneficial both from a technical and economic point of view [40]. However, the use of standard CNC systems (that follow a coded programmed instruction without operator interventions), does not mean the full automation of a machining process [41,42]. Nowadays, human operator supervision of each machine tool is still a necessity, no matter how modern the used CNC systems are. Therefore, it is a priority to solve the problem of the automation of machine tool supervision in the furniture industry. One of the most

important aspects of this supervision is tool condition monitoring. Therefore, autonomous drill condition monitoring systems are still problems waiting for effective solutions.

## 2. Fundamental Assumptions of the New Approach to Drill Condition Monitoring

One of the most innovative elements in the analyzed research trend is the adoption of the following four general methodological assumptions [33–38].

- Firstly—the traditional expectation that an effective tool condition monitoring system should be able to show the current value of a well-defined tool wear indicator (for example flank wear VB [26]) or, even better, estimate the working time until the tool needs to be changed [43] is absolutely exaggerated (excessive) from a practical point of view. The fully automated production systems need less sophisticated, yet much more clear suggestions: "Keep working—the tool is still able to cut" or "Stop working— the tool is worn out and must be replaced". Moreover, assuming the unavoidable (practically speaking) existence of an intermediate situation (on the border between "go on" and "stop"), the third type of message (warning message—for example: "Watch out—you can work for now, but check additional conditions for working capacity") was also found to be useful in industrial practice. On this basis, it was concluded that the current condition of the drill should be classified (analogous to traffic rules) as "Green" (which means that the drill is able to continue working), "Yellow" (which means a warning state), or "Red" (which means that the drill is unable to work). It is worth highlighting that this simplified point of view defines the drill condition monitoring problem as a standard three-class classification. This is a key advantage that creates the possibility of accepting the next assumption.
- Secondly—effective drill condition monitoring strategies can be based on an artificial intelligence algorithm commonly used for three-class classification. An important advantage of this approach is that there is no need to invent new strategies. Instead, it is enough to try well-known algorithms that have been proven effective in other areas of technology.
- Thirdly—drill condition monitoring systems do not have to be based only on traditional signals that are generated in real time during machining (and can be analyzed "on-line"), such as cutting forces, acoustic emission (in the audible and ultrasonic frequency band), or workpiece vibrations [33,35]. An alternative solution can be an off-line analysis of the hole quality [34,36–38]. An essential advantage of this approach is that the drill condition monitoring system could be integrated with the quality management system. It would be very convenient from the point of view of furniture producers.
- Fourthly—the time series structure of the obtained data can be ignored (does not have to be analyzed). This assumption has a significant advantage—it facilitates the using and testing of standard algorithms commonly intended for three-class classification. Unfortunately, it also has controversial consequences. One of them is analyzing the data obtained while working with a tool that would normally be previously replaced with a new one. Therefore, it is worth emphasizing that this assumption is not in line with industrial standards and practice.

The first three of these assumptions seem quite reasonable. The fourth may be controversial from a practical point of view but seems to be acceptable at the research stage. This is because scientists creating the drill condition monitoring system (unlike the regular user of this system) can analyze all the data they want. Even those that would have never had a chance to be analyzed in a real industrial process.

## 3. Materials and Methods

All experiments were carried out under industrial-like conditions, with industrial CNC machine tools (Figure 1), standard tungsten carbide drill bits (Figure 2), and mass-produced melamine faced particleboards.

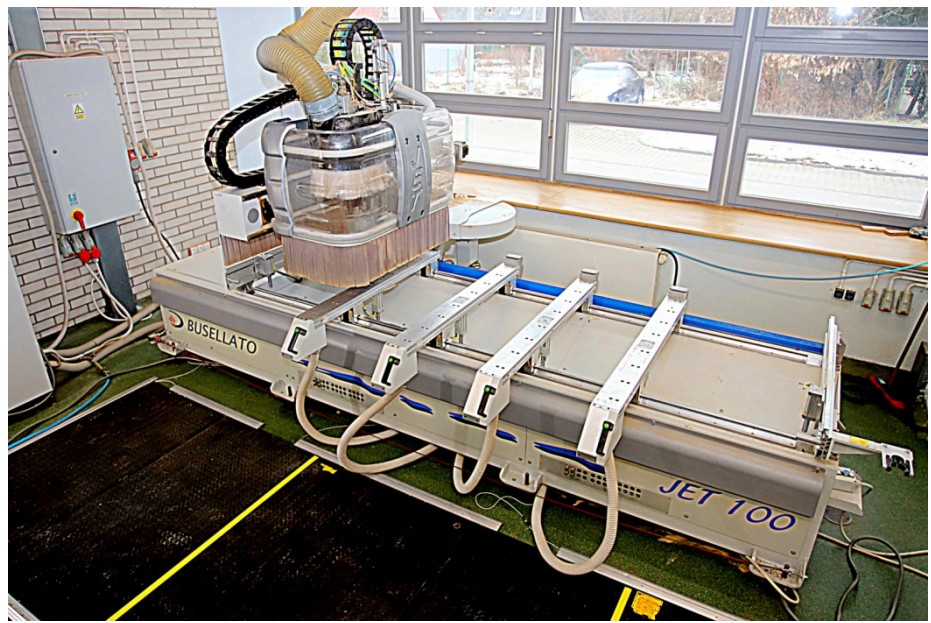

**Figure 1.** The standard CNC (computer numerical control) router (Busellato Jet 100), which was used during the experimental tests.

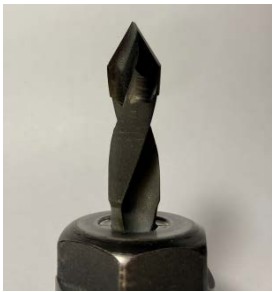

**Figure 2.** The standard two-blade drill bit intended for drilling in wood-based panels (catalogue symbol—K0500013, WP-01, FABA S.A. Poland), which was used during the experimental tests.

The tool wear was monitored manually in an objective way using workshop microscopes. The drill wear indicator was the size of outer corner wear denoted by the symbol "W" (Figure 3).

**Outer corner wear (W) = Margin width – Unworn margin**

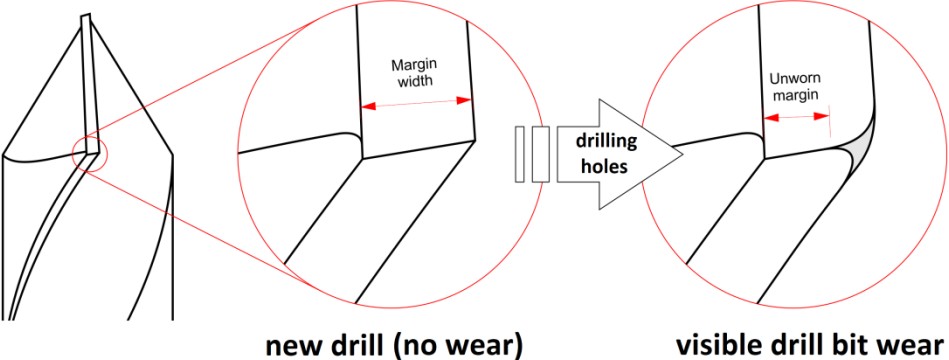

**Figure 3.** The way of outer corner wear (W) determination.

This basic indicator (W) was averaged for two drill bit blades and given in millimeters. Based on these measurements, the current state of the drill was manually and objectively classified (analogously to traffic rules) as "Green" (for W < 0.2 mm), "Yellow" (0.2 mm < W < 0.35 mm), or "Red" (W > 0.35 mm). In this way (according to the aforementioned, objective criterion), the real class was determined for each case. The real class means the target class for the drill condition monitoring system. The term target class is used because the tool wear monitoring system should "hit that target", that is, should classify the drill condition to the same class, however without human intervention (i.e., without viewing it under a microscope). In other words, the output (guessed) class of a perfect automatic monitoring system would always be the same as the target (real) class.

As mentioned in the previous chapter, two alternative concepts of the drill wear monitoring systems were developed and tested:

- Classification based on the automatic analysis of 5 digitalized signals generated in the machining zone (feed force, drilling torque, acceleration of jig vibration, audible noise, and ultrasonic acoustic emission) [33,35];
- Classification based on the automatic analysis of digital images of the drilled holes [34, 36–38].

The experimental setup used for the measurement and digital recording of the afore-mentioned signals was performed using the NI LabView (National Instruments—Austin, TX, USA) environment and using data acquisition cards (the scheme of the system is shown in Figure 4 [35]). The measurement of cutting forces required the use of a special force gauge in which the workpieces were clamped (Figure 5). During tests without measuring cutting forces, the workpieces were clamped using a standard vacuum working table.

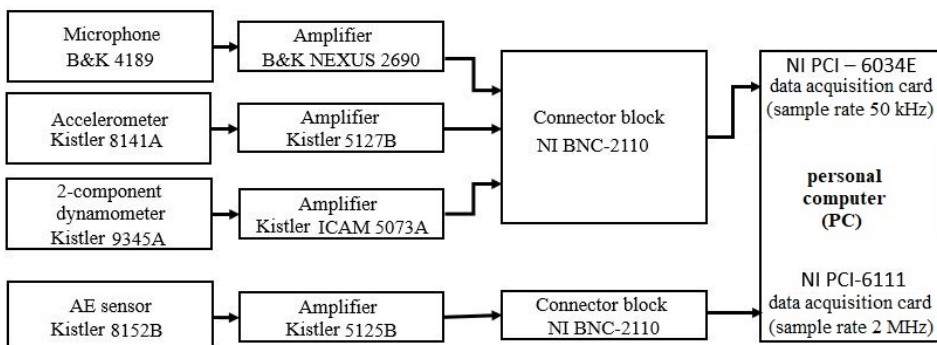

**Figure 4.** Scheme of the system used for measuring and digital recording of 5 selected signals generated during drilling: feed force, drilling torque, noise, ultrasonic acoustic emission (AE), and acceleration of jig vibration [35].

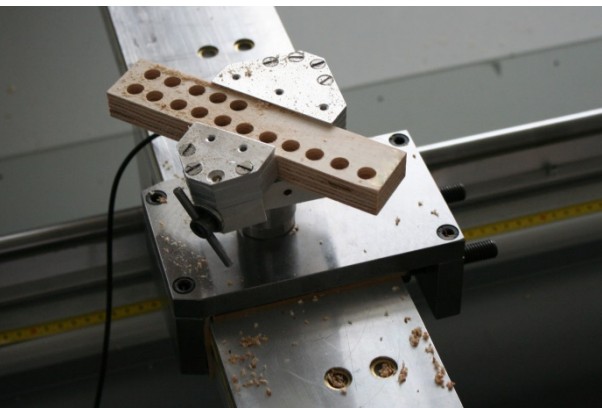

**Figure 5.** The special force gauge in which the workpieces were clamped.

The digital images of the drilled holes were obtained by scanning (1200 dpi) the workpieces using standard office scanners.

The fundamental factor determining the effectiveness of any classifier based on artificial intelligence is learning, not the classifier structure itself. Therefore, the following, standard way of learning and testing of the drill condition monitoring systems was adopted (Figure 6). The empirical data set was divided into two parts. One part was the training data set, which was used in the learning procedure to create a fully functional (ready-to-use) classifier. At the learning stage (learning mode), the classifier "knew" the real state of the monitored drill (drill wear indicator W). Next the classifier was tested. At the testing stage (testing mode), the task of the classifier was to classify the condition of the drills based only on indirect wear symptoms, such as signals measured according to Figure 4 or digital images of the drilled holes. The results of this classification were later compared with the real condition of the drill bits. The real condition of each drill bit is contained in the empirical data set.

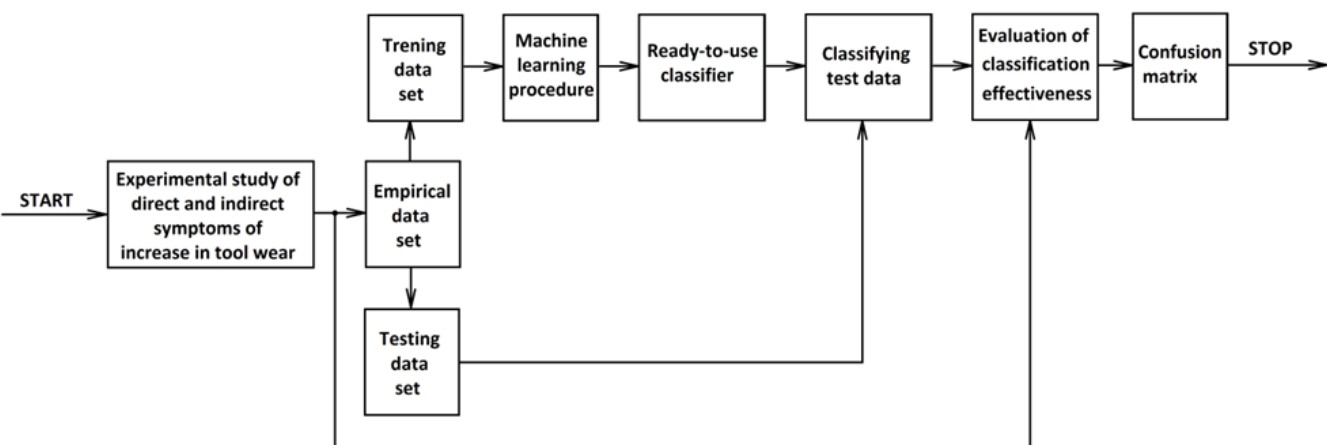

**Figure 6.** Block diagram of learning and testing the system of drill condition monitoring.

When analyzing the effectiveness of the various strategies of the drill condition monitoring systems, standard confusion matrices can be used. This type of matrix is the most common and clear (in the field of machine learning) way of visualization of the performance of any multi-class classifiers. The uniform confusion matrix structure used in the article is shown in Table 1.

**Table 1.** Uniform confusion matrix structure used in the article (additional explanations in the text).

| Output (Guessed) Class | "Green" | GG [%] | YG [%] | RG [%] |
|---|---|---|---|---|
| | "Yellow" | GY [%] | YY [%] | RY [%] |
| | "Red" | GR [%] | YR [%] | RR [%] |
| ACC [%] = GG [%] + YY [%] + RR [%] | | "Green" | "Yellow" | "Red" |
| CCE [%] = GR [%] + RG [%] | | | Target (Real) Class | |

Each matrix that will be shown later in the article has a percentage form (for easier comparison) and mostly consists of 11 percentage numbers that can be defined as follows.

- GG [%]—the percentage of observations calculated as the ratio of the count of true "Green" observations correctly assigned to the "Green" class to the total observation count;
- GY [%]—the percentage of observations calculated as the ratio of the count of true "Green" observations incorrectly assigned to the "Yellow" class to the total observation count;
- GR [%]—the percentage of observations calculated as the ratio of the count of true "Green" observations incorrectly assigned to the "Red" class to the total observation count;

- YG [%]—the percentage of observations calculated as the ratio of the count of true "Yellow" observations incorrectly assigned to the "Green" class to the total observation count;
- YY [%]—the percentage of observations calculated as the ratio of the count of true "Yellow" observations correctly assigned to the "Yellow" class to the total observation count;
- YR [%]—the percentage of observations calculated as the ratio of the count of true "Yellow" observations incorrectly assigned to the "Red" class to the total observation count;
- RG [%]—the percentage of observations calculated as the ratio of the count of true "Red" observations incorrectly assigned to the "Green" class to the total observation count;
- RY [%]—the percentage of observations calculated as the ratio of the count of true "Red" observations incorrectly assigned to the "Yellow" class to the total observation count;
- RR [%]—the percentage of observations calculated as the ratio of the count of true "Red" observations correctly assigned to the "Red" class to the total observation count;
- ACC [%]—the general classification accuracy in percentage form (a standard metric that summarizes the general effectiveness of a classifier), which were correctly classified, and can be calculated using the following formula:

$$ACC\ [\%] = GG\ [\%] + YY\ [\%] + RR\ [\%]; \tag{1}$$

- CCE [%]—the critical classification error (the percentage of observations, which were really "Red" however were identified by the monitoring system as "Green" or vice versa) which can be calculated using the following formula:

$$CCE\ [\%] = GR\ [\%] + RG\ [\%]. \tag{2}$$

## 4. Results and Discussion

A detailed overview of all the results of the scientific research obtained in the framework of the new approach to drill condition monitoring in wood-based panels machining [27–38] seems to be beyond the scope of the article. Besides, it is not necessary because all the original publications are widely available. Therefore, only a synthetic presentation has been made. Carefully selected data, which were taken from the most interesting papers, were processed in such a way as to create a coherent, systematic picture of the current state of the field of drill condition monitoring. The processed data makes it possible to directly compare the effectiveness of the various, latest monitoring concepts. For this purpose, the idea of confusion matrices was used.

Table 2 contains the best variants of confusion matrix that have been achieved in the five different studies belonging to the new (2019–2021) research trend in drill condition monitoring [33–38]. Different diagnostic data and different classification strategies (algorithms) were used. The general accuracy of classification (ACC [%]) ranged between 67% and 82%. The critical classification error (CCE [%]) ranged between 0% and 1.6%.

These results seem very promising, yet are still not good enough to develop a commercial monitoring system. The biggest issue lies in that even the advanced algorithms (e.g., Siamese networks or Deep learning) do not radically improve the general accuracy. Accuracy above 90% seems completely unattainable as of now. Under these circumstances, a completely new concept was adopted in the latest of the published studies pertaining to the analyzed research trend [38]. In short, it was decided to reduce the expectations that would be met by drill condition monitoring. More specifically, it has been found that the accuracy of the monitoring system can be increased by avoiding classifying the cases in which the system considers the most problematic (the most difficult to identify) and place them to the pseudo-class called "Sorry—I do not know what it is". It would form the fourth, additional class. Then, we had three standard classes: "Green", "Red", "Yellow", and one pseudo-class named "Unknown". In industrial practice, a human expert would deal with the further analysis of the cases placed in the "Unknown" pseudo-class. Such auto-limitation of the monitoring system seems much better (fairer) than the high risk of making completely incorrect classifications. Of course, this idea is not quite original and is normally recommended when the goal of automatic classification is to support rather than

replace a human expert. In case of drill condition monitoring, this would be a great step in the right direction, however not the final solution to the full automation problem.

**Table 2.** Confusion matrices that have been achieved in the five different studies belonging to the new research trend in drill condition monitoring [33–38].

| Reference No. | Diagnostic Data | Classification Strategy (Algorithm) | The Best Version of Matrix Confusion | | | |
|---|---|---|---|---|---|---|
| [33] | Digitalized signals of cutting forces, noise, acoustic emission, and acceleration of jig vibrations | Support vector machine (SVM) | Output (Guessed) Class — "Green" | 30.2% | 4.7% | 0% |
| | | | "Yellow" | 4.7% | 14.0% | 0% |
| | | | "Red" | 0% | 9.3% | 37.2% |
| | | | ACC = 81.4% CEE = 0% — "Green" | "Yellow" | "Red" | |
| | | | Target (Real) Class | | | |
| [34] | Digital images of drilled holes | Convolutional neural networks (CNN) | Output (Guessed) Class — "Green" | 38.6% | 5.7% | 0.2% |
| | | | "Yellow" | 4.5% | 24.6% | 4.1% |
| | | | "Red" | 0.1% | 8.1% | 14.1% |
| | | | ACC = 77.3% CEE = 0.3% — "Green" | "Yellow" | "Red" | |
| | | | Target (Real) Class | | | |
| [35] | Digitalized signals of cutting forces, noise, acoustic emission, and acceleration of jig vibrations | K-nearest neighbors (k-NN) | Output (Guessed) Class — "Green" | 29.8% | 6.5% | 0% |
| | | | "Yellow" | 5.1% | 11.2% | 1.9% |
| | | | "Red" | 0% | 10.2% | 35.3% |
| | | | ACC = 76.3% CCE = 0% — "Green" | "Yellow" | "Red" | |
| | | | Target (Real) Class | | | |
| [36] | Digital images of drilled holes | Siamese networks | Output (Guessed) Class — "Green" | 29.0% | 4.3% | 0.1% |
| | | | "Yellow" | 2.1% | 24.6% | 6.6% |
| | | | "Red" | 0.19% | 4.8% | 28.1% |
| | | | ACC = 81.8% CCE = 0.29% — "Green" | "Yellow" | "Red" | |
| | | | Target (Real) Class | | | |
| [37] | Digital images of drilled holes | Deep learning | Output (Guessed) Class — "Green" | 29.8% | 3.50% | 0% |
| | | | "Yellow" | 2.5% | 23.10% | 7.8% |
| | | | "Red" | 0.1% | 5.7% | 27.6% |
| | | | ACC = 80.5% CCE = 0.1% — "Green" | "Yellow" | "Red" | |
| | | | Target (Real) Class | | | |
| [38] | Digital images of drilled holes | Light gradient boosting machine (LGBM) | Output (Guessed) Class — "Green" | 35.9% | 7.8% | 0.7% |
| | | | "Yellow" | 6.8% | 18.5% | 7.5% |
| | | | "Red" | 0.9% | 9.3% | 12,6% |
| | | | ACC = 67.0% CEE = 1.6% — "Green" | "Yellow" | "Red" | |
| | | | Target (Real) Class | | | |

Table 3 contains the best variants of the confusion matrix that have been achieved in the most recent study pertaining to the new research trend in drill condition monitoring [38] for three different percentage of cases included in the "Unknown" pseudo-class: 0%, 40%, and 70%. It means that 100%, 60%, or only 30% of all case were classified respectively (this is how the "coverage" of observations was planned). The size of the pseudo-class was forced, however its content was determined by the monitoring system. The elimination of an increasing number of the most problematic cases (from 0% to 70%) resulted in a significant increase in classification accuracy (from 67% to 94.3%). This way, the level of accuracy was significantly exceeded by 90%, which was the goal. Unfortunately, this was achieved with only 30% coverage of observations (i.e., 70% of the cases would have to be analyzed by a machine tool operator in the traditional way). It seems that such a result cannot yet be the basis for developing a system that would be commercially attractive. This problem remains open and requires further research. Therefore, as already stated in the introduction, the main purpose of this review is to encourage more scientists to cooperate or compete in this research area now.

**Table 3.** Confusion matrices that have been achieved in the latest of the published studies pertaining to the new research trend in drill condition monitoring [38] for a different forced size of the "Unknown" pseudo-class.

| Diagnostic Data | Classification Strategy (Algorithm) | Percentage of Cases Included in the "Unknown" Pseudo-Class | The Best Version of Matrix Confusion | | | |
|---|---|---|---|---|---|---|
| Digital images of drilled holes | Light gradient boosting machine (LGBM) | 0% | Output (Guessed) Class | "Green" | 35.9% | 7.8% | 0.7% |
| | | | | "Yellow" | 6.8% | 18.5% | 7.5% |
| | | | | "Red" | 0.9% | 9.3% | 12.6% |
| | | | ACC = 67,0% CEE = 1,6% | "Green" | "Yellow" | "Red" |
| | | | | Target (Real) Class | | |
| | | 40% | Output (Guessed) Class | "Green" | 55.1% | 3.5% | 0.3% |
| | | | | "Yellow" | 6.4% | 8.7% | 6.5% |
| | | | | "Red" | 0.6% | 4.1% | 14.9% |
| | | | ACC = 78.6% CCE = 0.9% | "Green" | "Yellow" | "Red" |
| | | | | Target (Real) Class | | |
| | | 70% | Output (Guessed) Class | "Green" | 88.1% | 0.1% | 0.0% |
| | | | | "Yellow" | 3.4% | 0.1% | 2.0% |
| | | | | "Red" | 0.1% | 0.0% | 6.1% |
| | | | ACC = 94,3% CCE = 0,1% | "Green" | "Yellow" | "Red" |
| | | | | Target (Real) Class | | |

## 5. Conclusions

- The main aim of all analyses presented in the article was to show the current state of art on drill condition monitoring systems intended for furniture industry. The coherent and systematic review of the latest scientific developments in this field has been made. It turned out that the general effectiveness of the tested monitoring systems (accuracy of classification ACC [%]) ranged between 67% and 82%. The critical classification error (CCE [%]) ranged between 0% and 1.6%. These results seem very promising, yet are still not good enough to develop a commercial monitoring system.
- A more useful form of obtaining diagnostic data and more effective classification strategies are likely required. In consequence, the problem of drill condition monitoring remains open and requires further research studies. Therefore, as already stated in the introduction, the main purpose of this review is to encourage more scientists to cooperate or compete in the aforementioned research area right now.
- Regarding future research it appears that, firstly, new measuring sensors and new physical symptoms of drill wear must be sought as even advanced artificial intelligence algorithms are not effective without reliable empirical data.

**Funding:** This research received no external funding.

**Data Availability Statement:** The data presented in this study are available on request from the author.

**Conflicts of Interest:** The author declares no conflict of interest.

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
