# Peer review of "The Review of New Scientific Developments in Drilling in Wood-Based Panels with Particular Emphasis on the Latest Research Trends in Drill Condition Monitoring"

_forests, doi:10.3390/f13020242_

Round 1
Reviewer 1 Report
I had the opportunity to review the paper proposed for Forests Journal entitled "The Review of New Scientific Developments in Drilling in Wood-Based Panels with Particular Emphasis on the Latest Re-search Trends in Drill Condition Monitoring". The article concerns the monitoring of the wear of cutting edges of drills. This is very interesting and topical topic for machining processes, especially for wood and wood-based materials machining processes. However, I have a several comments, which i my opinion should be take into account in revision process of manuscript.
1) The title of the articles should be changed. As it stands, the title suggests that this is a review article, while the article itself presents some research results. The article did not review the state-of-art in a given issue, only a analysis of the drill bit evaluation system based on selected methods of monitoring the condition of the drill bit has been carried out. My suggestion for the title is e.g.: "Analysis of Evaluation System based on Selected Methods for Drill Condition Monitoring".
2) In Chapter 1 (Introduction), the rationale for monitoring the condition of the tool (drill bit) was discussed at great length, however, the ways in which the condition of the tool can be monitored during process execution were very laconically discussed. What should be supplemented.
3) In the summary of the first two chapters, which provide a sort of introduction, there is no formulation of the aims of the analyses carried out.
4) In the first part of the paper, as the cutting edge wear parameters are mentioned, among others VB (wear on the clearance/flank surface). It would be advisable to provide an illustrative draw of a drill bit with the described cutting edge geometry together with a representation of the parameter VB in question.
5) Methodology. The author mentions that standard CNC machine tools and standard drills were used for experimental research. Both machine tools and drills come in many types on the market. Therefore, it is necessary to specify specifically which machine tool was used, what kinematic parameters it had. It is also necessary to state specifically what type of drill bit was used in the research. Specify the geometry of the cutting edges and other important parameters of the cutting tool. It is best to illustrate the drill bit on a drawing or photo with the parameters entered. The specific parameters of the cutting process that were analysed should also be presented. It is also necessary to state the material which has been subjected to the cutting process.
6) On what basis was the wear size of the outer corner of the drill bit selected as the wear indicator? It is not one of the standard indicators used to describe the wear of a cutting edge. What were the reasons for this choice? How does the indicator relate to standard indicators?
7) The method of assessing the quality of a hole through the image of its outline on the main plane is not accurate enough. It only informs us if we have problems when starting the drilling process. However, tools can already generate insufficient hole quality on internal (cylindrical) surfaces.
8) Author presents a number of parameters to assess the condition of the drill bit. There is no information on how these parameters are correlated with actual cutting tool wear. When the cutting tool is in the "Green" category, at which acceptable wear parameters, cutting forces, etc. ?
9) How the values of the different parameters for matrix were determined. It is not clear to the reader and therefore the description of the results obtained is not clear for reader.
I my opinion the article needs significant corrections and additions, especially in the methodological part.
Reviewer 2 Report
The author nicely explained about recent developments in wood based panels of drilling condition monitoring. However, they need to add some of the following content to improver further,
As it is scientific development in drilling, need to use any of the techniques to represent the review conclusions.
Add graphical representations where ever possible so reader will attracted.
Reviewer 3 Report
The manuscript has some information for authors, but needs improvements for consideration.
- The manuscript must discusses the more recent trends in condition monitoring with respect to the wood machining.
- The materials and methods should discuss the different processes of wood machining technique inline with condition monitoring.
- The authors should disclose a simple flow chart indicating the wood machining condition monitoring.
- More detailed analysis on tools and methods are required.
- The results and discussion needs more improvements.
- Add the following references which related to wood machining.
Modeling for prediction of surface roughness in drilling MDF panels using response surface methodology
Journal of Composite Materials 45 (16), 1639-1646.
Optimization of delamination factor in drilling medium-density fiberboards (MDF) using desirability-based approach
The International Journal of Advanced Manufacturing Technology 45 (3-4), 370-381
Evaluation of surface roughness parameters (Ra, Rz) in drilling of MDF composite panel using Box-Behnken experimental design (BBD)
Int. J. Des. Manuf. Technol 5 (1), 52-62
Thrust force evaluation in drilling medium density fibre (MDF) panels using design of experiments
International journal of manufacturing technology and management 25 (1-3 …
- Pin-point the conclusions and add futuure recommendations.
Round 2
Reviewer 1 Report
Dear Author,
Thank you very much for your extensive responses with regard to the comments made during the review of the manuscript. Generally many of these responses are satisfactory, however not all. Information on machine tools and cutting tools and how they are holded in the spindle is very important. The system of Machine Tool - Workpiece - Chuck - Cutting Tool has a significant effect on the stiffness of the process and therefore on the quality of the workmanship and on the quality of the measurements that monitor the cutting process. Therefore, information on the specific MT-W-C-CT system used in the data acquisition process should be included in the article.
Similarly, if cutting blade geometry in terms of cutting edge wear is discussed, and parameters for assessing cutting edge wear are described, these should be identified for the reader in the form of a clear and readable drawing. This is essential to make the article transparent and readable for the reader.
Reviewer 3 Report
---
Author Response
Thank you very much.